# Functional Hyperconnectivity during a Stories Listening Task in Magnetoencephalography Is Associated with Language Gains for Children Born Extremely Preterm

**DOI:** 10.3390/brainsci11101271

**Published:** 2021-09-26

**Authors:** Maria E. Barnes-Davis, Hisako Fujiwara, Georgina Drury, Stephanie L. Merhar, Nehal A. Parikh, Darren S. Kadis

**Affiliations:** 1Perinatal Institute, Cincinnati Children’s Hospital Medical Center, Cincinnati, OH 45229, USA; gdrury@wayne.edu (G.D.); stephanie.merhar@cchmc.org (S.L.M.); nehal.parikh@cchmc.org (N.A.P.); 2Department of Pediatrics, University of Cincinnati College of Medicine, Cincinnati, OH 45229, USA; 3Division of Neurology, Cincinnati Children’s Hospital Medical Center, Cincinnati, OH 45229, USA; hisako.fujiwara@cchmc.org; 4Pediatric Neuroimaging Research Consortium, Cincinnati Children’s Hospital Medical Center, Cincinnati, OH 45229, USA; 5Neurosciences and Mental Health, Hospital for Sick Children, Toronto, ON M5G 0A4, Canada; darren.kadis@sickkids.ca; 6Department of Physiology, University of Toronto, Toronto, ON M5S 1A8, Canada

**Keywords:** prematurity, language, magnetoencephalography, magnetic resonance imaging, connectivity, development

## Abstract

Extreme prematurity (EPT, <28 weeks gestation) is associated with language problems. We previously reported hyperconnectivity in EPT children versus term children (TC) using magnetoencephalography (MEG). Here, we aim to ascertain whether functional hyperconnectivity is a marker of language resiliency for EPT children, validating our earlier work with a distinct sample of contemporary well-performing EPT and preterm children with history of language delay (EPT-HLD). A total of 58 children (17 EPT, 9 EPT-HLD, and 32 TC) participated in stories listening during MEG and functional magnetic resonance imaging (fMRI) at 4–6 years. We compared connectivity in EPT and EPT-HLD, investigating relationships with language over time. We measured fMRI activation during stories listening and parcellated the activation map to obtain “nodes” for MEG connectivity analysis. There were no significant group differences in age, sex, race, ethnicity, parental education, income, language scores, or language representation on fMRI. MEG functional connectivity (weighted phase lag index) was significantly different between groups. Preterm children had increased connectivity, replicating our earlier work. EPT and EPT-HLD had hyperconnectivity versus TC at 24–26 Hz, with EPT-HLD exhibiting greatest connectivity. Network strength correlated with change in standardized scores from 2 years to 4–6 years of age, suggesting hyperconnectivity is a marker of advancing language development.

## 1. Introduction

Prematurity impacts approximately 10% of births globally [1,2]. The limit of viability is shifting lower, with resuscitative care occurring as early as 22 weeks [3]. Some of these “periviable” infants are surviving beyond discharge from the neonatal intensive care unit (NICU). Extremely preterm (EPT, less than 28 weeks completed gestation) children may experience a range of neurodevelopmental impairments (NDI), including hearing and visual impairment, motor impairment such as cerebral palsy (CP), cognitive impairment, attentional and behavioral issues, and language impairment [4,5,6,7,8,9,10]. Historically, prognostic tools and large clinical trials have focused on rates of—and mechanisms underlying—moderate to profound NDI, with improvements in survival outpacing improvements in neurodevelopmental outcomes [11,12]. Factors that confer developmental risk for EPT children are well-described, but factors conferring resiliency are not well known [12,13,14,15,16,17]. Investigation into brain-based markers of resiliency is key, as recent reports suggest that a significant proportion of periviable children are performing within their expected grade level in school [18]. Furthermore, socioenvironmental factors such as maternal education might confer resiliency by lessening the association between brain injury noted in the NICU and later neurodevelopment [19].

Language development is of particular importance for children born EPT, not only due to its vital role in learning and cognition but also due to the special role it has in quality of life and formation of relationships with caregivers and peers [19]. EPT are at significant risk for language delay, with approximately 1 in 3 diagnosed with language impairment [5,6,20,21,22]. Standard tools, such as structural brain imaging at term-equivalent age and standardized language assessment at 2 years of age, fail to adequately predict later language functioning in preterm children [20,23]. Functional imaging allows assessment of representation and connectivity of specific brain networks, such as those supporting language comprehension and expression. Task-based and resting-state functional magnetic resonance imaging (fMRI) have been employed to investigate brain connectivity and language in former preterm children, reporting increased interhemispheric connectivity and increased involvement of right perisylvian cortex [24,25,26,27,28,29,30,31,32]. Investigations by our lab and others have reported that EPT have increased network strength (the sum of all connection weights) in interhemispheric language networks, but relationships between network strength and language scores have been inconsistent [32,33,34]. 

Functional MRI has many strengths, such as valid and reproducible spatial maps in response to standard language tasks, but it lacks the temporal resolution to assess fast neuronal activity. Thus, magnetoencephalography (MEG) is highly complementary to fMRI. MEG measures magnetic fields generated by electrical currents in the brain, predominately from pyramidal cells in the cortex [35]. This affords investigators a sub-millisecond temporal resolution, allows direct measurement of fast neuronal activity, and is less susceptible to distortion and attenuation from skull, fontanel, and scalp differences than electroencephalography (EEG) [36]. Importantly, MEG is very amenable to pediatric testing, as it is more quiet and less intimidating to young children versus MRI [37,38]. Our group was the first to investigate functional and effective connectivity supporting language in EPT children using MEG [32]. In our pilot work, the bilateral temporal spatial representation of language networks in EPT children on fMRI was not significantly different from term children (TC). There were, however, striking differences in network dynamics. Well-performing EPT children (scoring within normal limits on language and cognitive testing at ages 4 to 6 years with no known brain injury or neurological deficit) demonstrated increased interhemispheric functional connectivity versus TC during passive stories listening in MEG. We obtained diffusion MRI data in the same subjects during the same session. Structural connectometry revealed increased connectivity in an extracallosal interhemispheric pathway involving bilateral temporal areas and the right cerebellum. Connectivity in this pathway was significantly related to performance for EPT children, but not for TC [39]. These findings suggest children born EPT engage atypical language pathways to perform comparably to term peers. Hyperconnectivity might represent a brain-based marker of resiliency in EPT children, which could be used to target interventions and predict later functioning. 

Our theoretical model is that cognition emerges from the coordinated activity of distributed groups of neurons and brain regions, and that language is a key component of cognition vulnerable to disruption by preterm birth [5,6,20,21,22,40]. In the last few years, our work has demonstrated that preterm birth does not appear to impact the cortical representation of this network (regions activated during language tasks), but it does seem to impact the way in which that network functions (network dynamics). If a child is doing well following preterm birth, our work suggests that we can expect increased connectivity in that network as a way to compensate for the differential conditions imparted by spending the last trimester of gestation ex utero [32,34,39]. The aim of this report is to determine if hyperconnectivity is a robust marker of resiliency for preterm children to validate our earlier pilot work with a distinct and larger sample of contemporary EPT children. Additionally, we aim to further investigate how connectivity profiles might differ in a more heterogenous sample, specifically including those with a history of formally diagnosed language delay, deficit, or impairment (EPT-HLD) but without other neurological diagnoses or known brain injury, and to assess the degree to which hyperconnectivity relates to language attainments over time. To that end, we tested three hypotheses. (1) All extremely preterm children (EPT + EPT-HLD) would exhibit similar language representation as TC on fMRI, but would have increased interhemispheric functional connectivity on MEG as compared to TC. (2) Children born extremely preterm with a history of language delay or deficit (EPT-HLD) would have lower scores on language assessments than extremely preterm children without delay (EPT). (3) EPT-HLD children would have language topography similar to children born extremely preterm without delay (EPT), but connectivity and dynamics of the network would be significantly different between these two subgroups, with the EPT-HLD group exhibiting less functional hyperconnectivity than the EPT group.

## 2. Materials and Methods

### 2.1. Participants

In this observational study, we enrolled 58 children (eligible age range from 4 years 0 months to 6 years 11 months) from the greater Cincinnati area. Term children (TC, n = 32) were recruited using community research announcements at local hospitals, health awareness events, and pediatrician clinics. EPT children with (*n* = 9) and without (*n* = 17) a history of a formal diagnosis of language delay or deficit were recruited from the Neonatal Research Network (NRN) Low Birth Weight Follow-Up Study, and through a query of billing codes at NICUs (level 3 or 4) in Cincinnati identifying children who met our exclusion and inclusion criteria (Table 1). We expected, based on the known incidence of prematurity and 2015 Vermont Oxford Network data (network-wide and locally), for our cohort of extremely preterm children to have approximately 50% males. We also expect a race/ethnicity distribution of 60–70% White/Caucasian, 25–35% Black/African–American, 2–6% Asian/Pacific Islander, and 7–24% Hispanic/Latino. No participant was excluded based on race, ethnicity, or sex. We anticipated our group of term children would have a sex, race, and ethnicity distribution similar to that of the greater Cincinnati area. The enrolled sample was congruent with US Census data in terms of the distribution of variables such as sex, race, and ethnicity, and children performed within normative limits on standardized assessments (Table 2). Extremely preterm children were eligible if born at <28 weeks gestation and without parenchymal lesions, hemorrhage, periventricular leukomalacia, or interventricular hemorrhage above grade 2 on neonatal cranial ultrasound. TC were included if they were born between 37 and 42 weeks. If a child had a history of language delay (defined for this study as current or prior formal diagnosis by pediatrician and/or speech language pathologist of language delay, deficit, disorder, or impairment in the medical record or history of speech/language therapy for such diagnosis) and/or had received speech and language therapy, they were excluded from the TC group. However, EPT children with a history of language delay or deficit were included in EPT-HLD group provided they met other inclusion and exclusion criteria (Table 1). Children with diagnoses of cerebral palsy, seizures, migraines, other neurological or psychiatric disorders, or learning disabilities were excluded from all groups.

For the purposes of this study, language delay was defined as a formal diagnosis of language delay or deficit (at any time prior to the study visit) in the medical chart by a pediatrician or speech language pathologist. Seven of the nine children in the EPT-HLD group had a diagnosis of language delay, language deficit, language disorder, or language impairment placed in the medical chart by pediatrician or neonatologist at follow up clinic. The most common diagnosis was “mixed receptive-expressive language disorder”. These 7 children were referred to SLP, where diagnosis was confirmed and speech and language therapy was initiated. The range of ages at diagnosis was 1 year 3 months to 2 years 3 months. Two of the nine children in the EPT-HLD group received medical care outside of our system, but were noted by the parent to have been in speech therapy due to language delay or deficit at 2 years of age. Children who had a history of feeding/swallowing therapy or pure articulation deficit (such as lisp or stutter) were excluded from all groups (TC, EPT, and EPT-HLD).

This study was approved by the Institutional Review Boards of Cincinnati Children’s Hospital Medical Center and TriHealth Perinatal Scientific Review Committee. This study conforms to the US Federal Policy for the Protection of Human Subjects and was carried out in accordance with the Declaration of Helsinki. We obtained written informed consent from parents and legal guardians and verbal assent from the children participating. All structural scans were read by a clinical pediatric neuroradiologist. Four TC and 1 EPT child were excluded after the neuroradiologist noted clinically significant incidental findings on their structural MRI scans (Chiari malformations, concern for focal injury/mass). These 5 children are not included in the above numbers or in our final analyses. Assessments were completed during a single visit for all but 1 participant and required a total of 4 hours of participation. 

### 2.2. Demographic and Neuropsychological Assessments at 4 to 6 Years

Parents completed MRI screening forms with study staff, as well as questionnaires regarding birth history, demographics, and socioeconomic items including parental education level and family income. While the parent/legal guardian was completing the forms, the child participated in standardized neuropsychological assessments, including the Peabody Picture Vocabulary Test (PPVT4) [41]; Expressive Vocabulary Test (EVT2) [42]; the Wechsler Nonverbal Scale of Ability (WNV) [43], and the Word Structure subtest of the Clinical Evaluation of Language Fundamentals: Preschool Edition (CELF-P) [44]. The EVT2 and PPVT4 were used to assess expressive and receptive vocabulary, respectively. Both the EVT2 and PPVT4 have high reliability and content validity; correlate highly with verbal intelligence, especially in children [45,46]; and have been used in several studies of preterm children [24,47,48,49,50]. The WNV was designed to use with groups at high risk of language impairment or groups for whom English might not be the primary language and served as our measure of general abilities. The Word Structure subtest of the CELF-P was used to assess a non-vocabulary measure of language (specifically morphometry and pragmatics).

### 2.3. Retrospective Extraction of Language Scores from 2 Years Corrected Age

A subset of our preterm participants (7 EPT-HLD and 12 EPT) had previously been evaluated at approximately 2 years corrected age with the Bayley Scales of Infant Development (BSID) 3rd Edition, a widely used assessment in longitudinal studies of prematurity [51]. These scores had been obtained as part of other ongoing longitudinal studies at CCHMC. Standardized scores from the Language scale are included in this study.

### 2.4. Stories Listening Task

For fMRI and MEG, acquisition occurred while children participated in a passive stories listening task. This paradigm has been extensively used, consisting of 5 stories developed at our center by a speech language pathologist, targeted for this age range, and presented bi-aurally in a female voice [32,52]. Between stories, children listened to speech-shaped noise, matched to the story stimuli for duration, spectral content, and amplitude envelope. The task was approximately 6 minutes in duration.

### 2.5. Magnetic Resonance Acquisition at 4 to 6 Years

#### 2.5.1. Structural MRI Acquisition

All MR scanning was conducted on a Philips Achieva 3.0T scanner. 3D T1 weighted structural images had 1.0 × 1.0 × 1.0 mm isotropic voxels with a 256 × 256 resolution matrix. T1 images had TR/TE = 8.055 ms/3. Structural scans lasted 5 min each. Only T1 magnetic resonance images were used here for construction of head models.

#### 2.5.2. Functional MRI Acquisition

fMRI recordings involved multi-echo acquisition (TE 14/32/50ms, TR 1226.45 ms), acquired with multiband (factor 3) and in-plane SENSE (factor 3) acceleration. The multi-echo acquisition provides greater signal-to-noise and the ability to isolate BOLD signal through independent component analysis [53,54]. Functional imaging voxels were 3.0 × 3.0 × 3.0 mm.

### 2.6. Magnetoencephalography Acquisition at 4 to 6 Years

MEG data were always acquired before MRI to prevent potential magnetization effects. Data were obtained with a 275-channel whole-head CTF system (MEG International Services Ltd., Coquitlam, BC, USA) at 1200 Hz sampling rate. Subjects were tested while supine, listening to stimuli via a calibrated audio system (Etymotic Research, Elk Grove Village, IL, USA). Head localization coils were placed at nasion and preauricular locations to monitor movement continuously. Following acquisition, radio-opaque markers were placed over the fiducial positions, to facilitate co-registration with structural MRI. The entire MEG session lasted approximately 30 min.

### 2.7. Analysis of Demographic and Neuropsychological Data

Between groups, differences were evaluated using ANOVA for continuous variables (age at time of testing, gestational age at birth, and neuropsychological test scores) and Fisher’s exact test for categorical variables (sex, race, ethnicity, and parental education and income). Performance metrics were then related to functional connectivity metrics using bivariate correlations.

### 2.8. Processing of Magnetic Resonance Data

Multi-echo fMRI data were processed using an automated independent component analysis pipeline, designed to isolate BOLD signal [53,54]. The process involved T2* weighted averaging of echoes followed by denoising—only components showing echo-dependence (i.e., BOLD signal) were retained. The conditioned data were conventionally analyzed in a GLM framework, using SPM12 (http://www.fil.ion.ucl.ac.uk/spm/software/spm12, last accessed: 16 November 2017) running in MATLAB R2020a (https://www.mathworks.com/products/matlab.html, last accessed 29 July 2020). In brief, the conventional analysis pipeline involved co-registration of structural MRI to the functional images, normalization to template space (MNI 152), and smoothing with a full width half maximum (FWHM) of 5 mm. As part of the multi echo ICA, the 4D images are spatially aligned (rigid body) prior to component estimation. As such, there is no residual movement in the 4D data set that is analyzed conventionally in SPM, so the conventional processing pipeline did not require motion correction. Contrast maps were generated (stories minus noise) for each subject and passed on to second-level analyses. A one-way ANOVA was performed in SPM12 to assess differences in language representation (clusters of voxel-wise activation) corrected for multiple comparisons (family wise error *p* < 0.05). There were no statistically significant differences between groups. To objectively identify the language network in our cohort, we then computed the joint activation map across groups (as there were no group differences in representation detected). For this joint activation map, we included 19 preterm participants and randomly selected 19 of the TC participants to ensure equal representation and a balanced joint activation map. The activation map was parcellated using a 200-unit random parcellation scheme [55]. Centroids of parcels with significant activation (greater than 10% active voxels) served as nodes for subsequent MEG connectivity analyses [32,56].

### 2.9. Processing of Magnetoencephalography Data

MEG data were analyzed using FieldTrip, an open-source MATLAB toolbox [57]. Line noise was attenuated at 60, 120, 180 Hz by means of a sharp discrete Fourier transform filter; the data were then bandpass filtered from 0.1 to 100 Hz. Each trial epoch included 0 to 2000 ms from onset of stories versus noise stimuli. Data were subjected to automatic jump artifact detection and contaminated trials were rejected. Realistic head models were constructed from each participant’s 3D T1 images [58]. Using a linearly constrained minimum-variance beamformer (LCMV) we estimated the time series of activity at each network node (virtual sensor analysis) similar to our previously reported work [32].

### 2.10. Functional Connectivity Analyses

Weighted phase lag index (wPLI) is a measure of phase difference distribution across trials. Consistent phase differences are reflected in greater wPLI values, indicating nontrivial functional connectivity between a pair of nodes [59]. Frequency analysis was performed on MEG data extracted from virtual sensors, using a Fourier transform with a DPSS taper from 2 to 70 Hz +/− 4 Hz smoothing. We calculated wPLI across trials and visualized the connectivity spectra for all groups to assess differences between all preterm children and TC. We investigated differences among all three groups (TC, EPT-HLD, and EPT) using a one-way ANOVA. We noted a contiguous frequency band of statistical significance on the ANOVA from 24–26 Hz. We then investigated post hoc comparisons and group differences in network extent within that frequency band using Network Based Statistics (NBS) [60]. We assessed differences at a range of initial t thresholds with 5000 permutations and family wise error correction of *p* < 0.05.

### 2.11. Graph Theoretical Analyses

To determine if MEG connectivity at 4–6 years was related to cognitive task performance, we computed total network strength (sum of absolute debiased wPLI for all pairwise connections) and assessed correlation with standardized assessment scores [40]. 

### 2.12. Analyses of Sex as a Biological Variable

Due to the known differential effect of sex on prematurity and language outcomes, sex was included in analyses as a biological variable [61,62,63]. Group differences in sex distribution and differences in outcomes such as language scores and network strength for females versus males were investigated.

## 3. Results

### 3.1. Demographic and Neuropsychological Assessment

There were no significant group differences in age at testing, sex, race, ethnicity, maternal education, family income, or language scores (*p* > 0.05, Table 2). For the TC group, the mean age was 5.55 years, with 15 males participating. The EPT group had a mean age of 5.34 years and 7 males participating. Similarly, the EPT-HLD group had a mean age of 5.8 years with 4 males participating. Average PPVT4 scores for the TC, EPT, and EPT-HLD groups were 111, 110, and 108, respectively (*p* = 0.837). Average EVT2 scores for the TC, EPT, and EPT-HLD groups were 108, 105, and 99, respectively (*p* = 0.208). Average Word Structure subscale scores from the CELF-P were 10, 9, and 9 for the TC, EPT, and EPT-HLD groups, respectively (*p* = 0.485). For the WNV, mean scores were 105, 103, and 98 for the TC, EPT, and EPT-HLD groups, respectively (*p* = 0.458). Parental education differences between groups were approaching the level of significance (*p* = 0.056).

For the subset of our preterm participants (7 EPT-HLD and 12 EPT) who had BSID Language scores from 2 years corrected age, mean standardized BSID Language Scale scores were 102.5 for the EPT group and 87.9 for the EPT-HLD group, though the difference was not statistically significant (t (17) = −1.85; *p* = 0.081). BSID Language scores at 2 years corrected age did not significantly correlate with neuropsychological scores at 4 to 6 years of age (*p* = 0.222 with PPVT; *p* = 0.165 with EVT; *p* = 0.740 with WNV; and *p* = 0.651 with CELF-P Word Structure Subscale). 

### 3.2. Language Representation on Magnetic Resonance Imaging

There were no significant differences between groups in activation in response to stories listening in fMRI (family wise error correction of *p* < 0.05 was used). All three groups exhibited the pattern expected for children in this age range: bilateral temporal activation in response to stories listening versus noise (Figure 1A). Additionally, there were no significant differences in representation between the TC group and a combined group of all preterm participants (EPT + EPT-HLD).

### 3.3. Functional Connectivity on Magnetoencephalography

After extraction of virtual sensors (visualized in Figure 1B with coordinates of the virtual sensors/nodes listed in Table 3) we assessed functional connectivity using MEG data. To relate findings to our previous pilot work, we investigated functional connectivity differences between the TC group and all preterm children (EPT + EPT-HLD) using an independent samples t-test (see Figure 2). Preterm children had significantly increased functional connectivity compared to TC in several frequency bands, including 14.5 to 15.5 Hz, 24.5–26.5 Hz, and 30–31.5 Hz. Functional connectivity between all three groups (TC, EPT-HLD, and EPT) was significantly different as assessed by a one-way ANOVA (*p* < 0.05). EPT and EPT-HLD showed functional hyperconnectivity versus TC at 24–26 Hz with EPT-HLD exhibiting the highest connectivity (see Figure 3). This frequency band was the focus of subsequent analyses.

Network Based Statistics (NBS) was used to identify significant subnetworks within the 24–26 Hz frequency range that could be contributing to the observed group differences in connectivity and to perform post hoc comparisons. A significant subnetwork was identified in which the combined preterm group (EPT-HLD + EPT) had greater connectivity than TC (*p* < 0.05 with family wise error correction, see Figure 4A) at a range of t thresholds (0.1–3.8). The interhemispheric network involved 30 nodes and 83 edges, including some in the cerebellum, similar to our previously reported work in another sample of EPT children [32].

Following the omnibus ANOVA, all 6 post hoc pairwise contrasts were evaluated. Of these, only two post hoc comparisons were significant. The first significant subnetwork was identified wherein network extent was greater for the EPT-HLD group than the TC group (*p* < 0.05) at a range of t thresholds from 1 to 5 (see Figure 4B for visualization of this subnetwork at a median t threshold of 3). This interhemispheric subnetwork involved 23 nodes and 16 edges. The second subnetwork (EPT-HLD > EPT) was significant at a range of t statistic thresholds (2.2–3.5, *p* < 0.05 with family wise error correction, see Figure 4C). This network is comprised of 13 nodes and 13 edges and is predominantly in the left hemisphere. Other contrasts, including EPT-HLD < EPT and EPT-HLD < TC, were not significant.

### 3.4. Network Strength and Relation to Performance

#### 3.4.1. All Extremely Preterm Children versus Term Children

Preterm children (EPT and EPT-HLD combined) exhibited significantly increased network strength versus their term counterparts (mean strength 38.11 versus 24.34, *p* < 0.05). However, this was not significantly correlated with any neurocognitive scores at 4 to 6 years of age across or within groups (*p* > 0.05).

#### 3.4.2. Extremely Preterm Children without History of Language Delay or Deficit (EPT)

Standardized neurocognitive assessment scores at 4–6 years of age were not significantly correlated with network strength in any of the identified subnetworks for the EPT group (*p* > 0.05).

#### 3.4.3. Extremely Preterm Children with History of Language Delay or Deficit (EPT-HLD)

Standardized neurocognitive assessment scores at 4–6 years of age were not significantly correlated with network strength in any of the identified subnetworks for the EPT-HLD group (*p* > 0.05).

#### 3.4.4. Sub-Analysis: Correlation with BSID Scores at 2 Years Corrected Age

As a sub-analysis, we investigated the relationship between BSID language scores at 2 years corrected age and strength of the significant subnetworks at 4–6 years of age. When all 17 preterm children with BSID were included, BSID scores negatively correlated with network strength across the EPT-HLD > TC subnetwork (rho = −0.57, *p* = 0.017) and across the EPT-HLD > EPT subnetwork (rho = −0.57, *p* = 0.018). When analyses were limited to the EPT group only (*n* = 12 with BSID scores), we failed to detect a correlation between strength in the significant subnetworks and BSID language scores at 2 years (*p* > 0.05). When analyses were limited to the EPT-HLD group only (*n* = 7 with BSID scores), BSID scores at 2 years were negatively correlated with network strength in all significant subnetworks, including the All Preterm > TC subnetwork (rho = −0.86, *p* = 0.013); the EPT-HLD > EPT subnetwork (rho = −0.84, *p* = 0.018); and the EPT-HLD > TC network (rho = −0.88, *p* = 0.01).

#### 3.4.5. Sub-Analysis: Correlation with Change in Standardized Language Scores

Contrary to our second hypothesis, our group of preterm children with a history of language delay (EPT-HLD) no longer had significant differences in scores as compared to EPT peers without history of language delay or compared to TC. In fact, all groups performed within normal limits on assessments at 4 to 6 years of age. To investigate possible gains in language functioning, we calculated a language difference score by subtracting BSID standardized language scores from PPVT standardized scores at 4–6 years of age and investigated the relationship between these difference scores and network strength for the significant subnetworks. Of our 26 preterm participants, 19 had language difference scores, with a mean difference of 14, indicating a positive change in z-score. Three children exhibited a negative change and 16 exhibited a positive change. Language difference scores were positively correlated with network strength for the EPT-HLD > TC subnetwork (rho = 0.63, *p* = 0.006) and the EPT-HLD > EPT subnetwork (rho = 0.62, *p* = 0.008) suggesting they account for 40% of the variance in network strength (n = 17 preterm children with BSID scores). When analyses were limited to the EPT-HLD group only (n = 7 EPT-HLD with BSID scores), this positive correlation between language difference scores and strength in the EPT-HLD > TC network increased (rho = 0.78) and remained statistically significant (*p* = 0.039) accounting for 60% of the variance. When analyses were limited to the EPT group only (n = 12 with BSID scores), the correlations were not significant (*p* > 0.05).

### 3.5. Analyses of Sex Effects

There were no significant differences in sex distribution across groups (Table 2). When sex was investigated as an independent variable, there were no significant differences between males and females in standardized neurocognitive scores or strength of significant brain networks. However, these findings should be interpreted with caution, as the small number of participants in subgroups by sex could weaken any conclusions drawn.

## 4. Discussion

The reported results demonstrate that functional hyperconnectivity is found in well-performing EPT children at school age, and that it might serve as a marker for resiliency in this population. We have identified a significant bitemporal interhemispheric subnetwork in preterm children versus TC, consistent with our previous work [32,34]. Network strength for this subnetwork was positively correlated with gains in standardized language scores from age 2 years to age 4–6 years for preterm children. This suggests that functional hyperconnectivity is a measure of preterm adaptation and resiliency.

These conclusions are supported by our experiments in a number of ways. We validated our previously reported finding of interhemispheric functional hyperconnectivity in EPT children despite no significant differences in representation on task-based fMRI (confirming our first hypothesis). This was replicated in a distinct, larger sample of EPT children despite using different fMRI acquisition parameters (including a multi-echo approach). Our preterm participants (EPT-HLD + EPT) exhibited interhemispheric hyperconnectivity involving bitemporal areas and cerebellar nodes, similar to the network we previously reported [32]. It is possible that the cerebellum, a structure undergoing remarkable development in the last trimester of gestation during which these preterm children were born, provides a stabilizing effect in the context of prematurity-associated dysmaturation which might preferentially involve periventricular areas [33,39,50,64]. The cerebellum is being increasingly recognized for its role in development of children born at term, including language and reading [65,66]. 

Our EPT-HLD participants were previously diagnosed with language delay, deficit, disorder, or impairment by a pediatrician and/or pediatric speech language pathologist. Our second hypothesis was that they would have lower standardized language scores than the EPT group, and that this would be related to decreased functional hyperconnectivity. This hypothesis proved incorrect. In fact, there were no significant differences in language scores between the EPT and EPT-HLD groups at 4 to 6 years of age.

Our third hypothesis was that EPT-HLD children would have language topography similar to children born extremely preterm children without delay (EPT), but that connectivity and dynamics of the network would be significantly different between these two subgroups, with the EPT-HLD group exhibiting less functional hyperconnectivity than the EPT group. This hypothesis was partially incorrect. The EPT-HLD group had statistically significantly increased functional connectivity compared to both the TC and EPT groups. We demonstrated significant connectivity differences between EPT children without a history of language delay and those EPT children with a history of language delay (EPT-HLD). Increased network strength within our data-driven fMRI-defined language network was positively correlated with observable change in language performance (though not with language scores at 4 to 6 years of age). For our subgroup of preterm children who had both BSID at 2 years corrected age (from participation in other studies at CCHMC) and language testing at 4 to 6 years of age, the difference (in either direction, although almost all children had higher scores at 4 to 6 years) in language scores accounted for 40% of the variance in network strength, suggesting that interhemispheric functional hyperconnectivity represents an adaptive response correlated with language gains in extremely preterm children specifically. 

Of note, all children in the EPT-HLD group had received formal speech/language therapy, and no child enrolled in the EPT or TC groups had received formal speech/language therapy. One could speculate that the observed hyperconnectivity in the setting of language scores that are within normal limits for the EPT-HLD group speaks to the effectiveness of speech and language therapies. However, we were unable to quantify the amount of therapy or characterize the kind of therapy for our participants. Even in situations in which the specific “brand” of therapy is known, it is often challenging to characterize the true nature of a given language therapy for a specific child. Parents only reported to study personnel if their child had ever been in speech or language therapy, which was confirmed by review of available medical records. This is an exciting avenue for future research, as the interhemispheric hyperconnectivity we report might serve as a marker for resiliency for EPT children specifically, or serve to index a response to speech and language therapy.

### 4.1. Limitations

This investigation has some limitations, the most significant of which is the small sample size, most notably the small number of children in the EPT-HLD group. Future studies will include larger samples of EPT-HLD children. Additionally, we had few participants for whom we had access to language testing results at 2 years corrected age. While mean BSID scores at 2 years corrected age for the EPT-HLD group were almost 1 standard deviation lower than the EPT group (and this is likely to be clinically meaningful), this difference was not statistically significant. We observed a negative correlation between BSID scores at 2 years corrected age and network strength for the EPT-HLD group. Interestingly, these children had normalization of performance by 4 to 6 years of age, and they experienced more gains than their EPT peers. We do not have imaging for these children at 2 years of age. We do not have a group of EPT children who failed to normalize language performance or who scored significantly lower on language assessments at 4–6 years. Comparison with such a group is needed to determine definitively that interhemispheric hyperconnectivity is a brain-based marker of resiliency in the context of extreme prematurity, versus a mere correlate of preterm birth. Despite this, network strength did significantly correlate with language gains, suggesting it is a marker for a good outcome despite the risks of prematurity.

Interpretation of the language difference scores could be seen as a limitation of the study. Ideally, such a score would be calculated using the same instruments or assessments that capture the same dimensions of language and cognition. We are using data from widely used assessments and trying to convey the results from that data as clearly and honestly as possible. We do believe that the BSID and the PPVT and EVT are—at a minimum—in the same cognate area, as they assay language skills at a gross level. Vocabulary is frequently used as a rapid test for verbal IQ (although it can be argued that verbal IQ captures many dimensions beyond simple lexicon). We are using assessments which test different (or at least not identical) aspects of language in development. We appreciate that this might make interpretability of the difference score more difficult, but we do not think it completely invalidates our interpretation. 

While there were no statistically significant differences between groups in terms of parental education, this result did approach significance (Table 2). There were more children in the TC group who had parents with high school as the highest level of education attained and more children in the TC group who had parents with post-graduate education. Given the significance of socioeconomic factors—including parental education—in the language development of preterm children, this finding warrants further investigation in larger studies [14,19,22,67]. 

Finally, our EPT-HLD group did not have statistically significantly lower language scores at 4–6 years of age compared to our EPT and TC groups, despite a formal diagnosis of language delay or deficit (first noted in the medical chart at 1 to 2 years of age). This is congruent with some literature suggesting language impairment in children born preterm improves with age, although there is conflicting evidence for the resolution or persistence of deficits [47,68,69]. We relied primarily on two vocabulary tests (the PPVT and EVT) which other investigators have noted might fail to capture the full extent of more complex language abilities or difficulties [70]. This is a limitation of the study. Our results suggest that these EPT-HLD children experienced a delay or deficit in language, but not a true language impairment. Future studies should investigate criteria commonly used by Neonatologists, Developmental Pediatricians, and Speech Language Pathologists in NICU follow-up settings to determine consistency and prognosis in such diagnoses.

### 4.2. Strengths

Our study is unique in that we combine multimodal imaging methods synergistically, harnessing the temporal resolution of MEG in our connectivity analyses and using task-based fMRI to identify nodes based on cortical activation during a language task, in contrast to previous reports relying on resting state fMRI connectivity [26,27,29,71]. Functionally important network dynamics might only emerge during task demands. Additionally, we include children from a narrow band of well-defined gestational ages and chronological ages who have been cared for in the current era of neonatology, including those that—at the time of their birth—might have been considered periviable. Following such children is paramount to improving the care and counseling we provide to them and to their families. Furthermore, we included a group of extremely preterm children who had been formally diagnosed with language delay or deficit and had received speech/language therapy with subsequent normalization of standardized scores from assessments of language and general abilities, representing a unique cohort followed by MEG and fMRI into school age. Future studies will continue to follow this cohort longitudinally. Finally, by focusing on brain-based markers of resiliency and longitudinal gains in language functioning, we are shifting scientific focus from a deficit-based to a strength-based approach, which is of value not only for EPT children but also their parents and providers.

### 4.3. Conclusions

This report provides evidence for the EPT brain having a plastic potential to reorganize in innovative ways, potentially compensating for known patterns of injury or dysmaturation after being born during a critical period of central nervous system development. Atypical network dynamics, such as our observed interhemispheric bitemporal functional hyperconnectivity, might serve as adaptive mechanisms in the context of prematurity, as they positively correlate with gains in standardized language scores enabling these extremely preterm children to perform comparably to term children. Additionally, the extremely preterm children with a history of language delay or deficit (EPT-HLD) at ages 2 to 3 years no longer had significant differences from the EPT children without a history of language delay or deficit at 4 to 6 years, and all children in the EPT-HLD group had received speech and language therapy. This is a line of scientific inquiry that should be pursued in future studies. While there were no significant differences between groups in parental education, some might view this finding as approaching significance. Larger studies in extremely preterm children are needed to clarify this finding. Finally, future investigations should prioritize resiliency and strength alongside risk, promoting public health by closing the gap between improvements we have made in survival for EPT children and improvements in their long-term neurodevelopmental outcome and quality of life. Through studies such as these reported experiments, we hope to change the scientific and social framework for discussions of prematurity and development from one of deficit and risk to one of actionable interventions.

## Figures and Tables

**Figure 1 brainsci-11-01271-f001:**
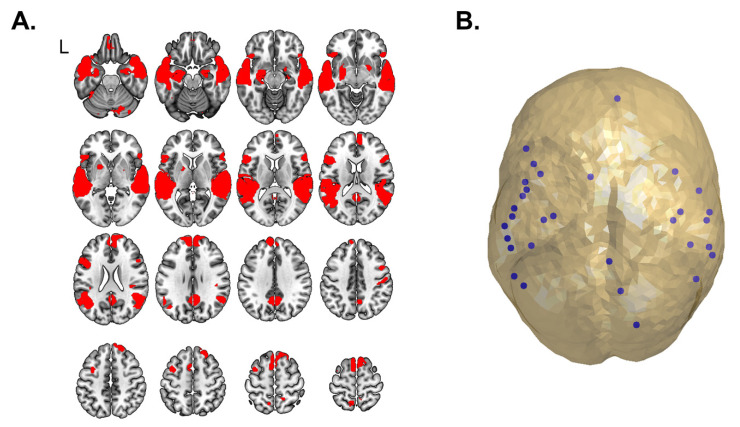
Joint functional MRI activation map and extraction of virtual sensors. (**A**) ANOVA was performed in SPM12 to assess differences in language representation (clusters of voxel-wise activation). There were no statistically significant differences between groups. To objectively identify the language network in our cohort, we then computed the joint activation map across groups (as there were no group differences in representation detected). Sixteen axial slices from the fMRI joint activation map (EPT-HLD + EPT + TC) are shown with typical bilateral activation in response to language stimuli (auditorily presented passive stories listening) versus noise condition (*p* < 0.001, k = 8). “L” denotes the left side of the brain in all images. (**B**) The joint activation map from fMRI was parcellated using a 200-unit random parcellation scheme. Centroids of parcels with significant activation (greater than 10% active voxels) served as “nodes” for subsequent connectivity analyses, shown in blue, that was performed on MEG data obtained during the same stories listening task.

**Figure 2 brainsci-11-01271-f002:**
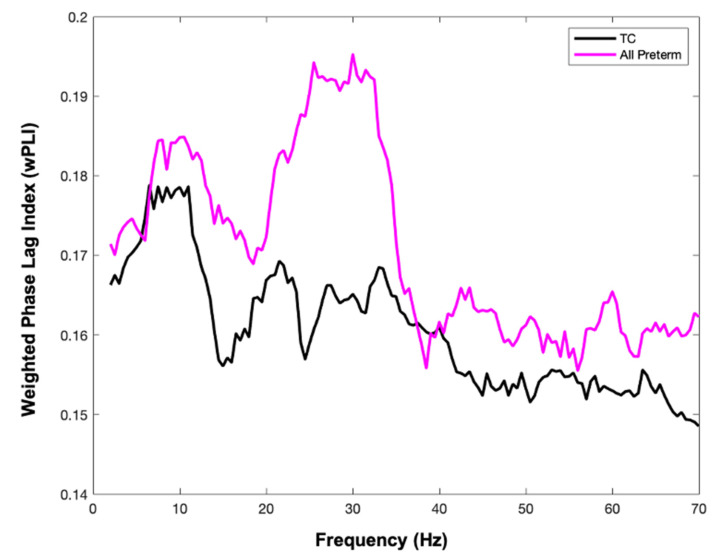
MEG functional connectivity indexed by weighted phase lag index for all preterm children and all term children. Weighted phase lag index (wPLI) extracted from timeseries at virtual sensors shown in Figure 1. All extremely preterm children (EPT-HLD + EPT, n = 26) are shown in pink, and term children (TC, n = 32) are shown in black. Statistically significant differences in connectivity between groups include increased functional connectivity for preterm participants at 14.5–15.5 Hz, 24.5–26.5 Hz, and 30–31.5 Hz. Images were generated in MATLAB (2020).

**Figure 3 brainsci-11-01271-f003:**
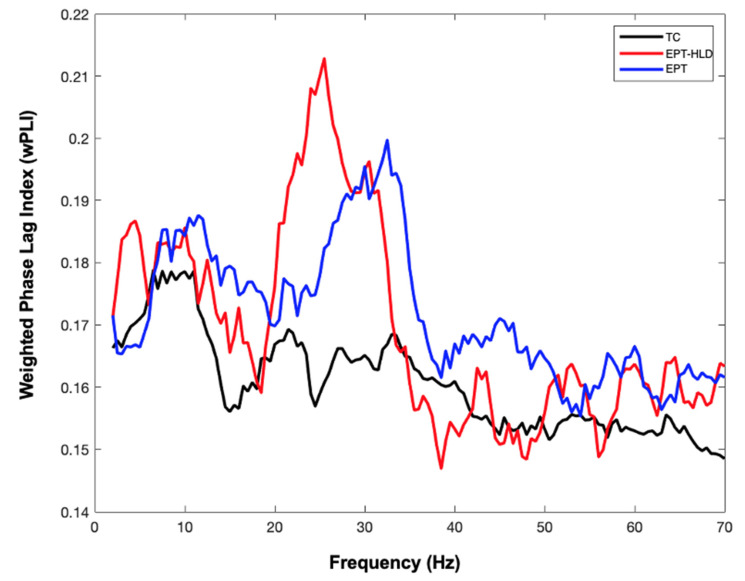
MEG functional connectivity indexed by weighted phase lag index for three groups: Extremely preterm with history of formally diagnosed language delay (EPT-HLD), extremely preterm without language delay (EPT), and term children (TC). Weighted phase lag index (wPLI) extracted from timeseries at virtual sensors shown in Figure 1. Functional connectivity between all three groups (TC in black, EPT-HLD in red, and EPT in blue) was significantly different as assessed by a one-way ANOVA (*p* < 0.05). EPT and EPT-HLD showed functional hyperconnectivity versus TC at 24–26 Hz with EPLI having highest functional connectivity. Images were generated in MATLAB (2020).

**Figure 4 brainsci-11-01271-f004:**
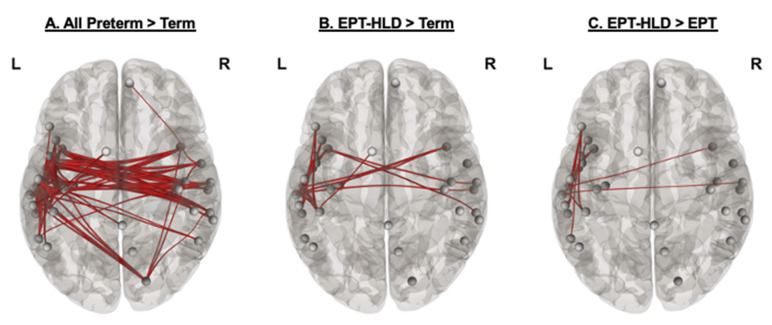
Significant subnetworks supporting hyperconnectivity in preterm children. Investigation of significant between groups differences (all preterm > TC) demonstrated a more bilateral network with the bulk of connections traversing the hemispheres. When significant differences were investigated between the EPT-HLD group and the EPT and TC groups in these post hoc comparisons, the significant connections appeared to be more concentrated in the left perisylvian region. “R” denotes the right side of the brain in all images. Images were generated from the adjacency matrices exported from NBS using the CONN toolbox running in MATLAB (2019b). (**A**) All extremely preterm children (EPT-HLD + EPT) collectively demonstrate significant subnetwork supporting hyperconnectivity versus term children (TC). Network “edges” showing significantly increased functional connectivity in all preterm participants versus TC between 24 and 26 Hz during stories listening (observed at various initial thresholds ranging from *t* =0.1 to 3.8, median t value of 1.95 shown, 5000 permutations, *p* < 0.05, corrected for multiple comparisons). (**B**) Extremely preterm children with a history of language delay (EPT-HLD) exhibit significant subnetwork supporting hyperconnectivity versus term children (TC). Network “edges” showing significantly increased functional connectivity in EPT-HLD versus TC between 24 and 26 Hz during stories listening (observed at various initial thresholds ranging from *t* =1 to 5, median t value of 3 shown, 5000 iterations, *p* < 0.05, corrected for multiple comparisons). (**C**) Extremely preterm children with a history of language delay (EPT-HLD) exhibit significant subnetwork supporting hyperconnectivity versus extremely preterm peer without language delay (EPT). Network “edges” showing significantly increased functional connectivity in EPT-HLD versus EPT between 24 and 26 Hz during stories listening (observed at various initial thresholds ranging from *t* =2.2 to 3.5, median t value of 2.9 shown, 5000 iterations, *p* < 0.05, corrected for multiple comparisons).

**Table 1 brainsci-11-01271-t001:** Inclusion and Exclusion Criteria.

**Term Control/Comparison Children (TC)**
Age 4 to less than 7 years
Personal history of term birth with gestational age of 37 weeks to 42 weeks
Informed consent of parent, assent of children
Negative for
Cerebral palsy
IVH Grade III or IV or parenchymal lesion/bleed on cranial ultrasound
Seizures
Migraines
History of speech, language, or learning disability
History of other neurologic or psychiatric disease, such as autism or ADHD
Standard MRI exclusion criteria, including orthodontic braces or metallic implants/devices
**Extremely Preterm Children Without Diagnosis of Language Impairment (EPT)**
Age 4 to less than 7 years
Personal history of preterm birth with gestational age of less than 28 weeks
Personal history of birth weight less than 1500 grams
Informed consent of parent, assent of children
Negative for
Cerebral palsy
IVH Grade III or IV or parenchymal lesion/bleed on cranial ultrasound
Seizures
Migraines
History of speech, language, or learning disability
History of other neurologic or psychiatric disease, such as autism or ADHD
Standard MRI exclusion criteria, including orthodontic braces or metallic implants/devices
**Extremely Preterm Children With History of Language Delay (EPT-HLD)**
Age 4 to less than 7 years
Personal history of preterm birth with gestational age of less than 28 weeks
Personal history of birth weight less than 1500 grams
Personal history of language delay or deficit
(Defined as current or prior formal diagnosis by pediatrician and/or speech language pathologist of language delay, deficit, disorder, or impairment in the medical record or history of speech/language therapy for such diagnosis)
Informed consent of parent, assent of children
Negative for
Cerebral palsy
IVH Grade III or IV or parenchymal lesion/bleed on cranial ultrasound
Seizures
Migraines
History of other neurologic or psychiatric disease, such as autism or ADHD
Standard MRI exclusion criteria, including orthodontic braces or metallic implants/devices

**Table 2 brainsci-11-01271-t002:** Demographics and Neuropsychological Data for Entire Sample.

		EPT-HLD (*n* = 9)	EPT (*n* = 17)	TC (*n* = 32)	*p* Value
Age (Years, Mean ± SD)		5.81 ± 0.64	5.34 ± 0.96	5.54 ± 0.95	0.47
Gestational Age (Weeks + Days)		25 + 5	26 + 3	39 + 3	<0.001
Sex	Females	5	10	17	0.935
	Males	4	7	15	
Race	White/Caucasian	4	11	20	0.727
	Black/African American	5	4	9	
	Other/Multiple	0	1	2	
	No Response	0	1	1	
Ethnicity	Hispanic/Latino/Latina	2	1	2	0.277
	Not Hispanic/Latino/Latina	7	16	30	
	No Response	0	0	0	
Family Income	<$50,000	4	4	11	0.886
	$50,000–$100,000	2	5	9	
	>$100,000	3	8	12	
	No Response	0	0	0	
Parental Education	High School	1	0	6	0.056
	College	6	10	9	
	Post Graduate	2	7	17	
	No Response	0	0	0	
Receptive Language	PPVT-4 (Mean ± SD)	108 ± 14	110 ± 12	111 ± 16	0.837
Expressive Language	EVT-2 (Mean ± SD)	99 ± 7	105 ± 12	108 ± 16	0.208
Language Morphology	CELFP-WS (Mean ± SD)	9.38 ± 2	9.38 ± 3	10.23 ± 3	0.485
General Abilities	WNV (Mean ± SD)	98 ± 14	103 ± 15	105 ± 17	0.458
Language Scores at Age 2	BSID3 (Mean ± SD)	87.9 ± 16	102.5 ± 17		0.081

Note: Categorical variables were tested using Fisher’s Exact Test and p values are reported. Continuous variables were tested using Analysis of Variance (ANOVA) tests and p values are reported. EPT-HLD = Extremely Preterm with History of Language Delay/Disorder. EPT = Extremely Preterm without Language Delay or Deficit. TC = Term Comparison Children. SD = Standard Deviation. PPVT-4 = Peabody Picture Vocabulary Test. EVT-2 = Expressive Vocabulary Test. CELFP-WS = Clinical Evaluation of Language Fundamentals Preschool Word Structure Scaled Score. WNV = Wechsler Non-Verbal Scale of Ability. BSID3 = Bayley Scales of Infant Development, 3rd Edition, Language Scaled Score.

**Table 3 brainsci-11-01271-t003:** Virtual Sensor Coordinates By Region.

	MNI Coordinates	Region
Left Frontal	−9, 7, 63	Left Superior Frontal
	−48, 25, 7	Left Inferior Frontal
Right Frontal	7, 55, 24	Right Medial Frontal
Left Temporal	−56, −12, 8	Left Primary Auditory
	−48, −1, −18	Left Middle Temporal
	−51, −4, −31	Left Inferior Temporal
	−62, −23, −17	Left Superior Temporal
	−49, 4, −2	Left Superior Temporal
	−60, −30, 13	Left Superior Temporal
	−44, 15, −16	Left Temporal Pole
	−57, −17, −17	Left Middle Temporal
	−38, −19, 13	Left Insula
	−32, −17, −21	Left Hippocampus
	−58, −36, −7	Left Middle Temporal
	−40, 9, −37	Left Middle Temporal
	−56, −53, 5	Left Middle Temporal
	−45, −35, 13	Left Superior Temporal
	−50, −59, 23	Left Superior Temporal
Right Temporal	62, −35, −8	Right Middle Temporal
	41, −19, 12	Right Primary Auditory
	45, −14, −7	Right Superior Temporal
	61, −19, −20	Right Inferior Temporal
	65, −40, −10	Right Superior Temporal
	56, −1, −22	Right Middle Temporal
	42, 11, −20	Right Superior Temporal
	63, −14, −1	Right Superior Temporal
	51, −34, 11	Right Superior Temporal
	55, −55, 21	Right Supramarginal
Right Parietal	9, −62, 33	Right Precuneus
	2, −44, 28	Right Posterior Cingulate
Cerebellar	18, −83, −30	Right Cerebellum

Note: MNI = Montreal Neurological Institute, Region defined by xjView v.10.0.

## Data Availability

All data are available from Barnes-Davis upon reasonable request.

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
