# Peer review of "Functional Hyperconnectivity during a Stories Listening Task in Magnetoencephalography Is Associated with Language Gains for Children Born Extremely Preterm"

_brainsci, 2021, doi:10.3390/brainsci11101271_

Round 1
Reviewer 1 Report
This is an interesting study on assessing the language brain network in children who were born extremely preterm and were studied with fMRI and MEG at the age of 4-6 years of age. The study examines three groups of children categorized as extremely preterm with a diagnosis of language impairment, extremely preterm with no diagnosis of language impairment, and age-matched healthy controls. By using a beamformer and functional connectivity analysis, the authors conclude that increased functional connectivity (hyperconnectivity) is associated with language gains for extremely preterm children.
The topic is relevant and interesting, the paper is well-written in most parts (except the discussion), the inclusion and exclusion criteria are well-stated, and neuroimaging experiments are by default hard to be performed in children of this age group.
Yet, the study has major and minor issues that should be addressed before the manuscript would be considered for publication.
- The title is misleading making the reader to believe that this is a neuroimaging study performed on preterm children. Please rephrase it by explicitly state that this is a study performed on children who were born extremely preterm.
- The introduction is well-written but quite long (particularly in comparison with the discussion section).
- In two occasions, the authors state that the main advantage of MEG (compared to functional MRI) is its higher temporal resolution. Although this statement is correct from a technical point of view, it fails to acknowledge the most important difference in the neurophysiological phenomena underlying these two measures. MEG measures directly the neural activity while fMRI the slower BOLD hemodynamic response that is an indirect measure of neural activity. Moreover, pediatric MEG presents significant advantages (compared to fMRI) for pediatric populations. You may want to refer and cite: Papadelis and Chen (Neuroimaging Clinics, 2020) and Chen et al. (Neuroimage, 2019).
- It is unclear to this reviewer why such a narrow age band was selected. Please explain the reason.
- Please state how co-operative were the children during the MEG and fMRI recordings. This is a difficult population to work with and movement related artifacts are a big challenge. For example, how frequent were movements during the MEG or fMRI. How much data of continuous MEG or fMRI with no movement artifacts were available for further analysis per group?
- Small cohort in order to consider any findings on sex differences as valid. Some sub-groups have 4 or 7 participants. Please remove these findings or indicate that their conclusions are weak.
- There is an "almost significant" difference between groups in terms of parental education (p=0.056). With an increase of the number of participants, this finding may reach statistical significance. This should be definitely discussed considering the relative literature.
- The figures are of poor quality. Please provide in figure 1 a full set of axial slides separately for each group in order the interested reader to appreciate the extend of the brain network. There is also a color-bar missing. At the same figure, it is unclear what the 1B depicts. How the location of virtual sensors was derived? Why they are unequal in terms of covering the two hemispheres? All these should be explained in a better way.
- The MEG findings in extremely short frequency bins (i.e., 14.5-15.5, 24.5-26.5, and 30-31.5 Hz) are definitely weak and may be eliminated as spurious from a more robust analysis. Why you do not perform an analysis across the different physiological EEG bands (e.g., alpha, beta1, beta2, gamma, etc)? On a similar note, why spectrograms in figures 2 and 3 end at 70 Hz? Filtering and analysis was performed up to 100 Hz. Do I miss something here?
- Please merge figures 4, 5, and 6 in one figure. Please display a color bar. It would be also interesting to compare the locations of the different nodes with regard to a brain atlas.
- The discussion is weak. It fails its most important purpose: to compare and discuss the presented findings with the existing literature. There is hardly 1-2 references listed in the discussion section. There are also some false statements that should be corrected: for example, lines 515-516 imply that task-based experiments can be performed only with the MEG. This is definitely not the case.
- I would eliminate strong statements from the discussion. See line 418: ...is a robust finding... The presented findings are (in the best case) weak to justify such an enthusiasm about their robustness.
Reviewer 2 Report
Thanks for recommending me as a reviewer. In this study, the authors were aim to ascertain whether functional hyperconnectivity is a marker of language resiliency for EPT children, validating our earlier work with a distinct sample of contemporary well-performing EPT and preterm children with history of language delay (EPT18 HLD). If the authors complete the revision, the quality of the study will be further improved.
- The introduction section is well written. If the authors describe the theoretical background of functional hyperconnectivity of language resiliency in more detail in the introduction section, it may help readers to understand.
2. line 103-108: Authors should be more specific about the general characteristics of the subjects (ex. sampling) in the Methods section.
3. Table 1: "Age 4.0 to less than 7 years" - In my opinion, the decimal point is not necessary.
4. It would be helpful to the reader if the authors add implications for future research in the Conclusion section.
